# Coresets for Wasserstein Distributionally Robust Optimization Problems

**Ruomin Huang**[1]     **Jiawei Huang**[2,3]     **Wenjie Liu**[2]     **Hu Ding**[*2]

[1]School of Data Science
[2]School of Computer Science and Technology
University of Science and Technology of China
[3]Department of Computer Science, City University of Hong Kong
`{hrm, hjw0330, lwj1217}@mail.ustc.edu.cn, huding@ustc.edu.cn`

## Abstract

Wasserstein distributionally robust optimization (WDRO) is a popular model to enhance the robustness of machine learning with ambiguous data. However, the complexity of WDRO can be prohibitive in practice since solving its "minimax" formulation requires a great amount of computation. Recently, several fast WDRO training algorithms for some specific machine learning tasks (e.g., logistic regression) have been developed. However, the research on designing efficient algorithms for general large-scale WDROs is still quite limited, to the best of our knowledge. *Coreset* is an important tool for compressing large dataset, and thus it has been widely applied to reduce the computational complexities for many optimization problems. In this paper, we introduce a unified framework to construct the $\epsilon$-coreset for the general WDRO problems. Though it is challenging to obtain a conventional coreset for WDRO due to the uncertainty issue of ambiguous data, we show that we can compute a "dual coreset" by using the strong duality property of WDRO. Also, the error introduced by the dual coreset can be theoretically guaranteed for the original WDRO objective. To construct the dual coreset, we propose a novel grid sampling approach that is particularly suitable for the dual formulation of WDRO. Finally, we implement our coreset approach and illustrate its effectiveness for several WDRO problems in the experiments. See arXiv:2210.04260 for the full version of this paper. The code is available at `https://github.com/h305142/WDRO_coreset`.

## 1 Introduction

In the past decades, a number of optimization techniques have been proposed for solving machine learning problems [49]. However, real-world optimization problems often suffer from the issue of data ambiguity that can be generated by natural data noise, potential adversarial attackers [3], or the constant changes of the underlying distribution (e.g., continual learning [42]). As a consequence, our obtained dataset usually cannot be fully trusted. Instead it is actually a perturbation of the true distribution. The recent studies have shown that even small perturbation can seriously destroy the final optimization result and could also yield unexpected error for the applications like classification and pattern recognition [20; 50].

The "**distributionally robust optimization** (DRO)" is an elegant model for solving the issue of ambiguous data. The idea follows from the intuition of game theory [41]. Roughly speaking, the

---

[*]Corresponding author.

36th Conference on Neural Information Processing Systems (NeurIPS 2022).

DRO aims to find a solution that is robust against the worst-case perturbation within a range of possible distributions. Given an empirical distribution $\mathbb{P}_n = \frac{1}{n} \sum_{i=1}^{n} \delta_{\xi_i}$ where $\delta_{\xi_i}$ is the Dirac point mass at the $i$-th data sample $\xi_i$, the **worst-case empirical risk** at the hypothesis $\theta$ is defined as $R^{\mathbb{P}_n}(\theta) = \sup_{\mathbb{Q} \in \mathcal{U}(\mathbb{P}_n)} \mathbb{E}^{\mathbb{Q}}[\ell(\theta, \xi)]$. Here $\mathcal{U}(\mathbb{P}_n)$ is the ambiguity set consisting of all possible distributions of interest, and $\ell(\cdot, \cdot)$ is the non-negative loss function. The DRO model has shown its promising advantage for enhancing the robustness for many practical machine learning problems, such as logistic regression [45], support vector machine [29], convex regression [5], neural networks [44; 48], etc.

In this paper, we consider one of the most representative DRO models that is defined by using optimal transportation [52]. **Wasserstein distance** is a popular measure for representing the difference between two distributions; it indicates the minimum cost for transporting one distribution to the other. For $p \geq 1$, the $p$-th order Wasserstein distance between two probability distributions $\mathbb{P}$ and $\mathbb{P}'$ supported on $\Xi$ is

$$W_p(\mathbb{P}, \mathbb{P}') = \left( \inf_{\pi \in \Pi(\mathbb{P}, \mathbb{P}')} \int_{\Xi \times \Xi} \mathsf{d}^p(\xi, \xi') \pi \, (\mathrm{d}\xi, \mathrm{d}\xi') \right)^{\frac{1}{p}}, \tag{1}$$

where $\mathsf{d}(\cdot, \cdot)$ is a metric on $\Xi$, and $\Pi(\mathbb{P}, \mathbb{P}')$ is the set of all joint probability distributions on $\Xi \times \Xi$ with the marginals $\mathbb{P}$ and $\mathbb{P}'$. By using the above Wasserstein distance (1), we can define the ambiguity set $\mathcal{U}(\mathbb{P}_n)$ to be the $p$-th order **Wasserstein ball** $\mathbb{B}_{\sigma,p}(\mathbb{P}_n)$, which covers all the distributions that have the $p$-th order Wasserstein distance at most $\sigma > 0$ to the given empirical distribution $\mathbb{P}_n$. The use of Wasserstein ball is a discrepancy-based approach for choosing the ambiguity set [41, section 5]. Also let

$$R_{\sigma,p}^{\mathbb{P}_n}(\theta) = \sup_{\mathbb{Q} \in \mathbb{B}_{\sigma,p}(\mathbb{P}_n)} \mathbb{E}^{\mathbb{Q}}[\ell(\theta, \xi)] \tag{2}$$

denote the corresponding worst-case empirical risk. The **Wasserstein distributionally robust optimization** (WDRO) problem [27] is to find the minimizer

$$\theta_* = \arg\min_{\theta \in \Theta} R_{\sigma,p}^{\mathbb{P}_n}(\theta) = \arg\min_{\theta \in \Theta} \sup_{\mathbb{Q} \in \mathbb{B}_{\sigma,p}(\mathbb{P}_n)} \mathbb{E}^{\mathbb{Q}}[\ell(\theta, \xi)], \tag{3}$$

where $\Theta$ is the feasible region in the hypothesis space. It is easy to see that the WDRO is a minimax optimization problem.

Compared with other robust optimization models, the WDRO model enjoys several significant benefits from the Wasserstein metric, especially for the applications in machine learning [55; 46; 6; 18]. The Wasserstein ball captures much richer information than the divergence-based discrepancies for the problems like pattern recognition and image retrieval [43; 33; 17]. It has also been proved that the WDRO model yields theoretical quality guarantees for the "out-of-sample" robustness [13].

However, due to the intractability of the the inner maximization problem (2), it is challenging to directly solve the minimax optimization problem (3). As shown in the work of Esfahani and Kuhn [13], the WDRO problem (3) usually has tractable reformulations [45; 35; 40; 29; 21; 4]. Although these reformulations are polynomial-time solvable, the off-the-shelf solvers can be costly for large-scale data. Another approach is to directly solve the minimization problem and the maximization problem alternatively [39] under a finite-support assumption. Gao and Kleywegt [17] proposed a routine to compute the finite structure of the worst-case distribution in theory. Nevertheless it still takes a high computational complexity if the Wasserstein ball has a large support size. Several fast WDRO training algorithms for some specific machine learning tasks, e.g., SVM and logistic regression by Li et al. [31, 32], have been developed recently; but it is unclear whether their methods can be generalized to solve other problems.

Therefore, it is urgent to develop efficient algorithmic techniques for reducing the computational complexity of the WDRO problems. **Coreset** is a popular tool for compressing large datasets, which was initially introduced by Agarwal et al. in computational geometry [1]. Intuitively, the coreset is an approximation of the original input data, but has a much smaller size. Thus any existing algorithm can run on the coreset instead and the computational complexity can be largely reduced. The coresets techniques have been widely applied for many optimization problems such as clustering and regression (we refer the reader to the recent surveys on coresets [36; 14]). Therefore a natural

idea is to consider applying the coreset technique to deal with large-scale WDRO problems. Below we introduce the formal definition of the coreset for WDRO problems.

**Definition 1 ($\epsilon$-coreset)** *Let $\epsilon$ be any given small number in $(0, 1)$. An $\epsilon$-coreset for the WDRO problem (3) is a sparse nonnegative mass vector $W = [w_1, \ldots, w_n]$, such that the total mass $\sum_{i=1}^{n} w_i = 1$ and the induced distribution $\tilde{\mathbb{P}}_n = \sum_{i=1}^{n} w_i \delta_{\xi_i}$ satisfies*

$$R_{\sigma,p}^{\tilde{\mathbb{P}}_n}(\theta) \in (1 \pm \epsilon) R_{\sigma,p}^{\mathbb{P}_n}(\theta), \forall \theta \in \Theta, \tag{4}$$

*where $R_{\sigma,p}^{\tilde{\mathbb{P}}_n}(\theta) := \sup_{\mathbb{Q} \in \mathbb{B}_{\sigma,p}(\tilde{\mathbb{P}}_n)} \mathbb{E}^{\mathbb{Q}}[\ell(\theta, \xi)]$ is the worst-case empirical risk of the coreset.*

It is worth to emphasize that the above coreset for WDRO is fundamentally different from the conventional coresets [14]. The main challenge for constructing the coreset of WDRO is from the "**uncertainty**" issue, that is, we have to consider all the possible distributions in the Wasserstein ball $\mathbb{B}_{\sigma,p}(\tilde{\mathbb{P}}_n)$; and more importantly, when the parameter vector $\theta$ varies, the distribution that achieves the worst-case empirical risk also changes inside $\mathbb{B}_{\sigma,p}(\tilde{\mathbb{P}}_n)$.

## 1.1 Our Contribution

In this paper, we propose a novel framework to construct the $\epsilon$-coresets for general WDRO problems. To the best of our knowledge, this is the first coreset algorithm for Wasserstein distributionally robust optimization problems. Our main contributions are twofold.

**-From coresets to dual coresets.** As mentioned before, it is challenging to directly construct the coresets for the WDRO problems. Our key observation is inspired by the strong duality property of the WDRO model [13; 4; 17]. We introduce the "dual coreset" for the dual formulation of the WDRO problems. We can neatly circumvent the "uncertainty" issue in Definition 1 through the dual form. Also, we prove that the dual coreset can yield a theoretically quality-guaranteed coreset as Definition 1.

**-How to compute the dual coresets.** Further, we provide a unified framework to construct the dual coresets efficiently. The sensitive-sampling based coreset framework usually needs to compute the "pseudo-dimension" of the objective function and the "sensitivities" of the data items, which can be very difficult to obtain [14] (the pseudo-dimension measures how complicated the objective function is, and the sensitivity of each data item indicates its importance to the whole input data set). Therefore we consider to apply the spatial partition approach that was initiated by Chen [9]; roughly speaking, we partition the space into a logarithmic number of regions, and take a uniform sample from each region. This partition approach needs to compute the exact value of the Moreau-Yosida regularization [38], which is a key part in the dual formulation of WDRO (the formal definition is shown in Proposition 1). However, this value is often hard to obtain for general $\Xi$ and general $\ell(\cdot, \cdot)$. For instance, suppose $\Xi$ admits a conic representation and the learning model is SVM, then computing the Moreau-Yosida regularization is equivalent to solving a convex conic programming [46, corollary 3.12]. For some machine learning problems, it is usually relatively easier to estimate the bounds of the Moreau-Yosida regularization[46, Theorem 3.30]. Based on this observation, we generalize the spatial partition idea and propose a more practical "grid sampling" framework. By using this framework, we only need to estimate the upper and lower bounds of the Moreau-Yosida regularization instead of the exact value. We also prove that a broad range of objective functions can be handled under this framework.

Due to the space limit, we leave the omitted proofs, discussions and experimental results to the full version of this paper [25].

## 1.2 Other Related Works

A number of coreset-based techniques have been studied before for solving robust optimization problems. For example, Mirzasoleiman et al. [34] designed an algorithm to generate coreset to approximate the Jacobian of a neural network so as to train against noisy labels. The outlier-resistant coresets were also studied for computing the robust center-based clustering problems [15; 16; 24; 12]. For the general continuous and bounded optimization problems [47], Wang et al. [53] proposed a

dynamic framework to compute the coresets resisting against outliers. Several other techniques also have been proposed for dealing with large-scale DRO problems, such as the PCA based dimension reduction methods [10; 11] and the stochastic gradient optimization methods [30; 37].

## 2 Preliminaries

We assume the input-output space $\Xi = \mathbb{X} \times \mathbb{Y}$ with $\mathbb{X} \subseteq \mathbb{R}^m$ and $\mathbb{Y} \subseteq \mathbb{R}$, and let $\mathcal{P}(\Xi)$ denote the set of Borel probability distributions supported on $\Xi$. For $1 \leq i \leq n$, each data sample is a random vector $\xi_i = (x_i, y_i)$ drawn from some underlying distribution $\mathbb{P} \in \mathcal{P}(\Xi)$. The empirical distribution $\mathbb{P}_n = \frac{1}{n} \sum_{i=1}^{n} \delta_{\xi_i}$ is induced by the dataset $\{\xi_1, \ldots, \xi_n\}$, where $\delta_{\xi_i}$ is the Dirac point mass at $\xi_i$. We endow $\Xi$ with the distance $\mathsf{d}(\xi_i, \xi_j) = \|x_i - x_j\| + \frac{\gamma}{2}|y_i - y_j|$, where $\|\cdot\|$ stands for an arbitrary norm of $\mathbb{R}^m$ and the positive parameter "$\gamma$" quantifies the transportation cost on the label. This distance function is used for defining the Wasserstein distance (1). We assume that $(\Xi, \mathsf{d})$ is a complete metric space.

In the rest of this paper, we consider the WDRO problems satisfying the following two assumptions. The first assumption is on the smoothness and boundedness of $\theta$. Similar assumptions have been widely adopted in the machine learning field [56; 53].

**Assumption 1 (Smoothness and Boundedness of $\theta$ [47])**

   (i) (Boundedness) *The feasible region $\Theta$ of the parameter space for the WDRO problem (3) is within a closed Euclidean ball $\mathbb{B}(\theta_{\mathrm{anc}}, l_{\mathrm{p}})$ centered at some "anchor" point $\theta_{\mathrm{anc}} \in \mathbb{R}^d$ with radius $l_{\mathrm{p}} > 0$;*

   (ii) (Lipschitz Smoothness[2]) *There exists a constant $L > 0$, such that for any $\xi \in \Xi$ and any $\theta_1, \theta_2 \in \mathbb{B}(\theta_{\mathrm{anc}}, l_{\mathrm{p}})$, we have $|\ell(\theta_1, \xi) - \ell(\theta_2, \xi)| \leq L \|\theta_1 - \theta_2\|_2$.*

The second assumption states that the loss function $\ell(\theta, \xi)$ is continuous and has a bounded growth rate on data $\xi$. The detailed growth rate functions are discussed in Section 5.

**Assumption 2 (Continuity and Bounded Growth Rate of $\xi$)**

   (i) (Continuity) *The loss function $\ell(\theta, \cdot)$ is continuous for any $\theta \in \Theta$;*

   (ii) (Bounded Growth Rate) *There exists some positive continuous growth rate function $\mathsf{C}(\theta)$ and $\xi_0 \in \Xi$ such that*
   $$\ell(\theta, \xi) \leq \mathsf{C}(\theta)\left(1 + \mathsf{d}^p(\xi, \xi_0)\right)$$
   *for any $\theta \in \Theta$ and any $\xi \in \Xi$.*

Now we state the strong duality for the WDRO, which is an important property to guarantee the correctness of our dual coreset method.

**Proposition 1 (Strong duality [13; 4; 17])** *For any upper semi-continuous $\ell(\theta, \cdot)$, any $\theta$ and any nominal distribution $\mathbb{P}$ with finite $p$-th moment, the worst-case risk satisfies*
$$R_{\sigma,p}^{\mathbb{P}}(\theta) = \inf_{\lambda \geq 0}\{\lambda \sigma^p + H^{\mathbb{P}}(\theta, \lambda)\}, \tag{5}$$

*where $H^{\mathbb{P}}(\theta, \lambda) := \mathbb{E}^{\mathbb{P}}[h(\theta, \lambda, \xi)]$ and $h(\theta, \lambda, \xi) = \sup_{\zeta \in \Xi}\{\ell(\theta, \zeta) - \lambda \mathsf{d}^p(\zeta, \xi)\}$ is the Moreau-Yosida regularization [38]. We use $\lambda_*^{\mathbb{P}}(\theta)$ to denote the $\lambda$ attaining the infimum in (5).*

**Remark 1** *By the definition of $h$, for any given $\theta \in \Theta$, we can deduce that there always exists some $\lambda_*^{\mathbb{P}}(\theta) < \infty$ attaining the infimum of (5).*

For the sake of convenience, we abbreviate $R_{\sigma,p}(\theta) = R_{\sigma,p}^{\mathbb{P}_n}(\theta)$, $H(\theta, \lambda) = H^{\mathbb{P}_n}(\theta, \lambda)$, $\lambda_*(\theta) = \lambda_*^{\mathbb{P}_n}(\theta)$, and $h_i(\theta, \lambda) = h(\theta, \lambda, \xi_i)$. We define the asymptotic growth rate function
$$\kappa(\theta) := \limsup_{\mathsf{d}(\xi, \xi_0) \to \infty} \frac{\ell(\theta, \xi) - \ell(\theta, \xi_0)}{\mathsf{d}^p(\xi, \xi_0)}$$

---

[2]The methods proposed in this paper can be easily extended to other types of smoothness, e.g., gradient Lipschitz continuity.

so as to conclude the continuity of $h_i(\cdot, \cdot)$ in the following two claims. Here $\xi_0$ is the point in Assumption 2 (ii).

**Claim 1 (Continuity of $h_i$ on $\theta$)** *For each $i \in \{1, \ldots, n\}$ and any fixed $\lambda \geq 0$, we have*
$$|h_i(\theta, \lambda) - h_i(\theta, \lambda')| \leq L\|\theta - \theta'\|_2,$$
*for any $\theta, \theta' \in \Theta$ with $\kappa(\theta), \kappa(\theta') \leq \lambda$.*

**Claim 2 (Continuity of $h_i$ on $\lambda$)** *For each $i \in \{1, \ldots, n\}$ and any fixed $\theta \in \mathbb{R}^d$, we have*
$$|h_i(\theta, \lambda) - h_i(\theta, \lambda')| \leq \max\{r_i^p(\theta, \lambda), r_i^p(\theta, \lambda')\}|\lambda - \lambda'|, \ \forall \lambda, \lambda' \geq \kappa(\theta),$$
*where $r_i(\theta, \lambda) := \min_{\zeta \in \Xi}\{\mathtt{d}(\zeta, \xi_i) \mid \ell(\theta, \zeta) - \lambda \mathtt{d}^p(\zeta, \xi_i) = h_i(\theta, \lambda)\}$ is the closest distance between $\xi_i$ and all the $\zeta$s that attain the supremum of $\ell(\theta, \zeta) - \lambda \mathtt{d}^p(\zeta, \xi_i)$ in $\Xi$.*

**Remark 2** *The reason that we let $\lambda \geq \kappa(\theta)$ in the above claims is that each $h_i(\theta, \lambda)$ goes to infinity if $\lambda < \kappa(\theta)$. Without loss of generality[3], we suppose $h_i(\theta, \kappa(\theta)) < \infty$ in this paper.*

# 3 From Coresets to Dual Coresets

In this section, we provide the concept of "dual coreset" and prove that it is sufficient to guarantee the correctness with respect to the WDRO coreset. First, we present the definition of the dual coreset via directly combining Proposition 1 and Definition 1. Suppose $I$ is an interval depending on $\theta$ (we will discuss this assumption in detail later).

**Definition 2 (Dual $\epsilon$-Coreset)** *A dual $\epsilon$-coreset for the WDRO problem (3) is a sparse non-negative mass vector $W = [w_1, \ldots, w_n]$ such that the total mass $\sum_{i=1}^{n} w_i = 1$ and*

$$\tilde{H}(\theta, \lambda) := \sum_{i=1}^{n} w_i h_i(\theta, \lambda) \in (1 \pm \epsilon)H(\theta, \lambda) \tag{6}$$

*for any $\theta \in \Theta$ and $\lambda \in I$.*

**Remark 3** *Note that we require the approximation guarantee holds not only for any $\theta \in \Theta$, but also for any $\lambda \in I$ in the above definition. This is also a key difference to the traditional coresets.*

By the discussion in Remark 2, we know $I \subset [\kappa(\theta), \infty)$. If we directly let $I = [\kappa(\theta), \infty)$, the dual coreset of Definition 2 requires to approximate the queries from all $\lambda \geq \kappa(\theta)$, which is too **strong** and can be even troublesome for the coreset construction. Below we show that a bounded $I$ is sufficient for guaranteeing a dual coreset to be a qualified WDRO coreset.

Given a non-negative mass vector $W = [w_1, \cdots, w_n]$, the corresponding weighted empirical distribution is $\tilde{\mathbb{P}}_n = \sum_{i=1}^{n} w_i \delta_{\xi_i}$. Recall that we define a parameter $\lambda_*^{\mathbb{P}}(\theta)$ for duality in Proposition 1. Together with Assumption 2, we show the boundedness of the $\lambda_*^{\tilde{\mathbb{P}}_n}(\theta)$ (abbreviated as $\tilde{\lambda}_*(\theta)$ for convenience) for $\tilde{\mathbb{P}}_n$. Let $[n] = \{1, 2, \cdots, n\}$. The following result is a key to relax the requirement for the dual coreset in Definition 2.

**Lemma 1 (Boundedness of $\tilde{\lambda}_*$)** *Given the empirical distribution $\mathbb{P}_n = \frac{1}{n}\sum_{i=1}^{n} \delta_{\xi_i}$, we define the value $\rho = \max_{i \in [n]}\{\mathtt{d}(\xi_i, \xi_0)\}$ that is the largest distance from the data samples to $\xi_0$. Here $\xi_0$ is defined in Assumption 2 (ii). For any $\theta \in \Theta$ and any mass vector $W$, the $\tilde{\lambda}_*(\theta)$ of the corresponding weighted empirical distribution $\tilde{\mathbb{P}}_n$ is **no larger than***

$$\mathtt{C}(\theta) \cdot \left(2^{p-1} + \frac{1 + 2^{p-1}\rho^p}{\sigma^p}\right), \tag{7}$$

*where $\mathtt{C}(\theta)$ is defined in Assumption 2 (ii). We use $\tau(\theta)$ to denote this upper bound $\mathtt{C}(\theta)\left(2^{p-1} + \frac{1 + 2^{p-1}\rho^p}{\sigma^p}\right)$.*

---

[3]It is possible that $h_i(\theta, \kappa(\theta)) = \infty$, e.g., $\ell(\theta, \xi)$ is the loss function of ordinary linear regression. In this case, the argument in this paper still holds with slight modification.

**Algorithm 1** Dual $\epsilon$-Coreset Construction

---

**Input:** The empirical distribution $\mathbb{P}_n = \frac{1}{n} \sum_{i=1}^{n} \delta_{\xi_i}$, the Lipschitz constant $L$, the "anchors" $\theta_{\text{anc}}$ and $\lambda_{\text{anc}}$, and corresponding radii $l_{\text{p}}$ and $l_{\text{d}}$; the parameter $\epsilon \in (0, 1)$; lower bound oracle $a_i(\cdot, \cdot)$ and upper bound oracle $b_i(\cdot, \cdot)$ for $i \in [n]$.

1. Compute $A = \frac{1}{n} \sum_{i=1}^{n} a_i(\theta_{\text{anc}}, \lambda_{\text{anc}})$ and $B = \frac{1}{n} \sum_{i=1}^{n} b_i(\theta_{\text{anc}}, \lambda_{\text{anc}})$.
2. Let $N = \lceil \log n \rceil$; initialize $W = [0, 0, \cdots, 0] \in \mathbb{R}^n$.
3. The dataset $\{\xi_1, \ldots, \xi_n\}$ is partitioned into $(N+1)^2$ cells $\{C_{ij} | 0 \le i, j \le N\}$ as (12).
4. For each $C_{ij} \ne \emptyset, 0 \le i, j \le N$:
   (a) take a sample $Q_{ij}$ from $C_{ij}$ uniformly at random, where the size $|Q_{ij}|$ depends on the parameters $\epsilon$, $l_{\text{p}}$, $l_{\text{d}}$ and $L$ (the exact value will be discussed in our following analysis in Section 4.2);
   (b) for each sample $\xi_k \in Q_{ij}$, assign the mass of quantity $w_k = \frac{|C_{ij}|}{n|Q_{ij}|}$;

**Output:** the mass vector $W = [w_1, w_2, \cdots, w_n]$ as the dual $\epsilon$-coreset.

---

**Remark 4** *(i) In practice, we usually normalize the dataset before training a machine learning model, which implies that $\rho$ is not large. (ii) It is worth noting that the above lemma can help us to compute an upper bound for $\lambda_*(\theta)$. For example, if letting $W = [\frac{1}{n}, \ldots, \frac{1}{n}]$, (7) directly yields an upper bound.*

The following theorem shows that the query region $I = [\kappa(\theta), \tau(\theta)]$ is sufficient for obtaining a coreset of the WDRO problem (3).

**Theorem 1 (Sufficiency of the bounded query region)** *If we let query region $I = [\kappa(\theta), \tau(\theta)]$ in Definition 2, the dual $\epsilon$-coreset defined in such way also satisfies the coreset of Definition 1.*

Therefore in the rest of this paper, we let $I = [\kappa(\theta), \tau(\theta)]$ in Definition 2. Theorem 1 also implies the following corollary. So we can only focus on solving the dual WDRO problem (3) on the obtained dual $\epsilon$-coreset.

**Corollary 1** *Given $\alpha \ge 1$, we suppose the parameter vector $\theta_0$ yields an $\alpha$-approximation obtained on the dual $\epsilon$-coreset. Then $\theta_0$ is also an $(\alpha \cdot \frac{1+\epsilon}{1-\epsilon})$-approximation of the original WDRO (3).*

To end this section, similar to $\mathbb{B}(\theta_{\text{anc}}, l_{\text{p}})$ in Assumption 2 (i), we define an interval $[\lambda_{\text{anc}} - l_{\text{d}}, \lambda_{\text{anc}} + l_{\text{d}}]$ centered at some "anchor" point $\lambda_{\text{anc}} > 0$ with radius $l_{\text{d}} > 0$. To ensure that $[\kappa(\theta), \tau(\theta)]$ is within the interval $[\lambda_{\text{anc}} - l_{\text{d}}, \lambda_{\text{anc}} + l_{\text{d}}]$ for all $\theta \in \Theta$, we let $\lambda_{\text{anc}} := \max_{\theta \in \Theta}\{\kappa(\theta_{\text{anc}}), \frac{\tau(\theta)}{2}\}$ and $l_{\text{d}} := \lambda_{\text{anc}}$.

## 4 The Construction of Dual Coresets

Following the results of Section 3, we show how to compute a qualified dual coreset in this section. Suppose we can evaluate the lower and upper bounds for each $h_i(\cdot, \cdot)$ with respect to a given couple $(\lambda_{\text{anc}}, \theta_{\text{anc}})$, namely, we have

$$a_i(\theta_{\text{anc}}, \lambda_{\text{anc}}) \le h_i(\theta_{\text{anc}}, \lambda_{\text{anc}}) \le b_i(\theta_{\text{anc}}, \lambda_{\text{anc}})$$

for $1 \le i \le n$. We defer the details for obtaining such upper and lower bounds for each application to Section 5.

### 4.1 The Construction Algorithm

We show the dual $\epsilon$-coreset construction procedure in Algorithm 1, where the high-level idea is based on the following grid sampling.

**Grid sampling.** Let $N = \lceil \log n \rceil$. Given the anchor $(\theta_{\text{anc}}, \lambda_{\text{anc}})$, we can conduct the partitions over the dataset based on the lower bounds $a_i(\theta_{\text{anc}}, \lambda_{\text{anc}})$ and upper bounds $b_i(\theta_{\text{anc}}, \lambda_{\text{anc}})$ separately. Let $A = \frac{1}{n} \sum_{i=1}^{n} a_i(\theta_{\text{anc}}, \lambda_{\text{anc}})$ and $B = \frac{1}{n} \sum_{i=1}^{n} b_i(\theta_{\text{anc}}, \lambda_{\text{anc}})$. Then we have the following partitions.

$$
\begin{align}
A_0 &= \{\xi_i \mid a_i(\theta_{\text{anc}}, \lambda_{\text{anc}}) \le A\}, \tag{8} \\
A_j &= \{\xi_i \mid 2^{j-1}A < a_i(\theta_{\text{anc}}, \lambda_{\text{anc}}) \le 2^j A\}, 1 \le j \le N. \tag{9}
\end{align}
$$

$$B_0 = \left\{ \xi_i \mid b_i(\theta_{\text{anc}}, \lambda_{\text{anc}}) \leq B \right\}, \tag{10}$$

$$B_j = \left\{ \xi_i \mid 2^{j-1}B < b_i(\theta_{\text{anc}}, \lambda_{\text{anc}}) \leq 2^j B \right\}, 1 \leq j \leq N. \tag{11}$$

We denote the lower bound and upper bound partitions as $\mathcal{A} = \{A_0, \cdots, A_N\}$ and $\mathcal{B} = \{B_0, \cdots, B_N\}$ respectively. Then, we compute the intersections over $\mathcal{A}$ and $\mathcal{B}$ to generate the "grid":

$$\mathcal{C} = \{C_{ij} | C_{ij} = A_i \cap B_j, 0 \leq i, j \leq N\}. \tag{12}$$

It is easy to see that $\mathcal{C}$ is a collection of disjoint "cells" and $\bigcup_{i,j} C_{ij} = P$. For each $\xi_k \in C_{ij}$, we have

$$\mu_i \cdot 2^{i-1}A \leq h_k(\theta_{\text{anc}}, \lambda_{\text{anc}}) \leq 2^j B \tag{13}$$

where $\mu_i = 0$ if $i = 0$ and $\mu_i = 1$ otherwise. Through the grid partition $\mathcal{C}$, we can take a set of samples $Q_{ij}$ from $C_{ij}$ uniformly at random, and assign the weight $\frac{|C_{ij}|}{n|Q_{ij}|}$ to each sample.

**Remark 5** *(i) The grid sampling is a variance reduction technique in the Monte-Carlo methods [19], since the grid partition is also a stratification for $h_k(\theta_{\text{anc}}, \lambda_{\text{anc}})$ as shown in (13). If we consider only the upper bounds $b_i(\theta_{\text{anc}}, \lambda_{\text{anc}})$ or the lower bounds $a_i(\theta_{\text{anc}}, \lambda_{\text{anc}})$, the obtained partition is not a valid stratification for $h_k(\theta_{\text{anc}}, \lambda_{\text{anc}})$. (ii) If we can obtain the exact value of $h_i(\theta_{\text{anc}}, \lambda_{\text{anc}})$, i.e., $a_i(\theta_{\text{anc}}, \lambda_{\text{anc}}) = b_i(\theta_{\text{anc}}, \lambda_{\text{anc}}) = h_i(\theta_{\text{anc}}, \lambda_{\text{anc}})$, then the grid partition is exactly the spatial partition that was studied before [9; 53].*

## 4.2 Theoretical Analysis

In this section we analyze the complexity of Algorithm 1 in theory. Recall that we define $r_i(\theta, \lambda) = \min_{\zeta \in \Xi} \{ \mathtt{d}(\zeta, \xi_i) \colon \ell(\theta, \zeta) - \lambda \mathtt{d}^p(\zeta, \xi_i) = h_i(\theta, \lambda) \}$ in Claim 2. The following theorem provides an asymptotic sample complexity of Algorithm 1. To state the theorem clearly, we define two notations $R := \max_{\substack{i \in [n] \\ \theta \in \Theta}} \{r_i^p(\kappa(\theta), \theta)\}$ and $H := \min_{\substack{\theta \in \Theta \\ \lambda \in [\lambda_{\text{anc}} - l_{\text{d}}, \lambda_{\text{anc}} + l_{\text{d}}]}} H(\theta, \lambda)$.

**Theorem 2** *Set $|Q_{ij}| = \tilde{O}\left( \left( B \cdot \frac{B + Ll_{\text{p}} + Rl_{\text{d}}}{AH} \right)^2 \cdot \frac{d}{\epsilon^2} \right)^4$ in the Algorithm 1. Then the returned $W$ is a qualified dual $\epsilon$-coreset with probability at least $1 - \frac{1}{n}$. The construction time is $O(n \cdot \mathtt{time}_{ab})$ where $\mathtt{time}_{ab}$ is the time complexity for computing the lower bound $a_i(\theta_{\text{anc}}, \lambda_{\text{anc}})$ and the upper bound $b_i(\theta_{\text{anc}}, \lambda_{\text{anc}})$ for each $h_i(\theta_{\text{anc}}, \lambda_{\text{anc}})$.*

**Remark 6** *Note that the value $H \geq \min_{\theta \in \Theta} \mathbb{E}^{\mathbb{P}_n} \ell(\theta, \xi)$, which should not be too small in practice since the loss function $\ell(\cdot, \cdot)$ usually contains positive penalty terms. The value of $R$ will be discussed in Section 5.*

We show the sketched proof of Theorem 2 below. Based on the continuity of $h_i(\cdot, \cdot)$ and the Hoeffding's inequality [22], for a fixed couple $(\theta, \lambda)$, we provide an upper bound on the sample complexity first. The bound ensures that the estimation for each cell $C_{ij}$ has a bounded deviation with high probability.

**Lemma 2** *Let $\delta$ be a given positive number. We fix a couple $(\theta, \lambda) \in \mathbb{B}(\theta_{\text{anc}}, l_{\text{p}}) \times [\kappa(\theta), \tau(\theta)]$ and take a uniform sample $Q_{ij}$ from $C_{ij}$ with the sample size*

$$|Q_{ij}| = O\left( (2^j B - \mu_i \cdot 2^{i-1}A + 2Ll_{\text{p}} + 2Rl_{\text{d}})^2 \delta^{-2} \log \frac{1}{\eta} \right). \tag{14}$$

*Then, we have the probability*

$$\mathtt{Prob}\left[ \left| \frac{1}{|Q_{ij}|} \sum_{\xi_k \in Q_{ij}} h_k(\theta, \lambda) - \frac{1}{|C_{ij}|} \sum_{\xi_k \in C_{ij}} h_k(\theta, \lambda) \right| \geq \delta \right] \leq \eta. \tag{15}$$

---

[4] $\tilde{O}(g) := O(g \cdot \mathtt{polylog}(\frac{nLl_{\text{p}}Rl_{\text{d}}}{\epsilon H}))$

We aggregate the deviations from all the cells to obtain the overall estimation error for the coreset. To guarantee the approximation quality of (6), we need to design a sufficiently small value of the deviation $\delta$ for each cell $C_{ij}$ under our grid partition framework.

**Lemma 3** *In Lemma 2, we set the deviation $\delta = \epsilon_1(2^{j-1} + 2^{i-1})A$ for $0 \leq i, j \leq N$. Then we have*

$$\text{Prob}\left[|\tilde{H}(\theta, \lambda) - H(\theta, \lambda)| \leq 3\epsilon_1 H(\theta_{\text{anc}}, \lambda_{\text{anc}})\right] \geq 1 - (N+1)^2\eta. \tag{16}$$

To generalize the result of Lemma 3 to the whole feasible region $\mathbb{B}(\theta_{\text{anc}}, l_{\text{p}}) \times [0, 2l_{\text{d}}]$, we apply the discretization idea. Imagine to generate the axis-parallel grid with side length $\frac{\epsilon_3 l_{\text{p}}}{\sqrt{d}} \times \epsilon_2 l_{\text{d}}$ inside $\mathbb{B}(\theta_{\text{anc}}, l_{\text{p}}) \times [0, 2l_{\text{d}}]$; the parameters $\epsilon_2$ and $\epsilon_3$ are two small numbers that will be determined in our following analysis. For each grid cell we arbitrarily take a $(\theta, \lambda)$ as its representative point. Let $G$ be the set of the selected representative points; it is easy to see the cardinality $|G| = \frac{1}{\epsilon_2} \cdot O(\frac{1}{\epsilon_3^d})$. Through taking the union bound over all $(\theta, \lambda) \in G$ for (16), we obtain the following Lemma 4.

**Lemma 4** *With probability at least $1 - (N+1)^2|G|\eta$, we have*

$$|\tilde{H}(\theta, \lambda) - H(\theta, \lambda)| \leq 3\epsilon_1 H(\theta_{\text{anc}}, \lambda_{\text{anc}}) \text{ for all } (\theta, \lambda) \in G. \tag{17}$$

By using the above lemmas, we are ready to prove Theorem 2.

*Proof.*(**of Theorem 2**) For any $(\theta, \lambda) \in \mathbb{B}(\lambda_{\text{anc}}, l_{\text{d}}) \times [\kappa(\theta), \tau(\theta)]$, we let $(\theta', \lambda') \in G$ be the representative point of the cell containing $(\theta, \lambda)$. Then we have $\|\theta - \theta'\|_2 \leq \epsilon_3 l_{\text{p}}$ and $|\lambda' - \lambda| \leq \epsilon_2 l_{\text{d}}$. Without loss of generality, we assume $\lambda' \geq \lambda$. By using the triangle inequality, we have

$$\begin{aligned}
&|h_k(\theta, \lambda) - h_k(\theta', \lambda')| \\
\leq& |h_k(\theta', \lambda') - h_k(\theta, \lambda')| + |h_k(\theta, \lambda) - h_k(\theta, \lambda')| \\
\leq& L\epsilon_3 l_{\text{p}} + R\epsilon_2 l_{\text{d}}. \qquad \text{(By Claim 2, Claim 1 and } \lambda' \geq \lambda \geq \kappa(\theta))
\end{aligned} \tag{18}$$

The above inequality implies

$$|H(\theta, \lambda) - H(\theta', \lambda')| \leq R\epsilon_2 l_{\text{d}} + L\epsilon_3 l_{\text{p}} \tag{19}$$

and

$$|\tilde{H}(\theta, \lambda) - \tilde{H}(\theta', \lambda')| \leq R\epsilon_2 l_{\text{d}} + L\epsilon_3 l_{\text{p}}. \tag{20}$$

Overall we have $|\tilde{H}(\theta, \lambda) - H(\theta, \lambda)|$

$$\begin{aligned}
\leq& \left|\tilde{H}(\theta, \lambda) - \tilde{H}(\theta', \lambda')\right| + \left|\tilde{H}(\theta', \lambda') - H(\theta', \lambda')\right| + |H(\theta', \lambda') - H(\theta, \lambda)| \\
\leq& 3\epsilon_1 H(\theta_{\text{anc}}, \lambda_{\text{anc}}) + 2 \times \left(R\epsilon_2 l_{\text{d}} + L\epsilon_3 l_{\text{p}}\right) \quad \text{(By Lemma 4, (19) and (20))}
\end{aligned} \tag{21}$$

By setting $\epsilon_1 = \frac{H\epsilon}{9B}$, $\epsilon_2 = \frac{H\epsilon}{6l_{\text{d}}R}$, $\epsilon_3 = \frac{H\epsilon}{6Ll_{\text{p}}}$ and $\eta = \frac{1}{n(N+1)^2|G|}$ and substituting them into (14), we obtain the sample complexity as stated in Theorem 2. $\qquad\square$

# 5 Applications

In this section, we show several WDRO problems that their complexities can be reduced by using our dual coreset method.

## 5.1 Binary Classification

For the binary classification, $\mathbb{Y} = \{-1, 1\}$ and the loss function $\ell(\theta, \xi) = L(y \cdot \theta^\top x)$ where $L(\cdot)$ is a non-negative and non-increasing function. Let $\|\cdot\|_*$ be the dual norm of $\|\cdot\|$ on $\mathbb{R}^m$. We consider the **Support Vector Machine (SVM)** and **logistic regression** problems. The SVM takes the hinge loss $L(z) = \max\{0, 1 - z\}$ and the logistic regression takes the logloss $L(z) = \log(1 + \exp(-z))$. If $\mathbb{X} = \mathbb{R}^m$ and $p = 1$, by the result of Shafieezadeh-Abadeh et al. [46, Theorem 3.11], for both of these two problems we have:

- $R \leq \gamma$, $\kappa(\theta) = \mathtt{C}(\theta) = \|\theta\|_*$;
- $a_i(\theta, \lambda) = b_i(\theta, \lambda) = h_i(\theta, \lambda) = \max\{L(y_i \cdot \theta^\top x_i), L(-y_i \cdot \theta^\top x_i) - \lambda\gamma\}$ for any $\lambda \geq \kappa(\theta)$.

If $\mathbb{X} = [0, l]^m$ is a $m$-dimensional hypercube with side length $l > 0$, $h_i(\theta, \lambda)$ in fact is the optimal objective value of a convex constrained programming problem. Therefore we **cannot** obtain the exact value of $h_i(\theta, \lambda)$ easily. To remedy this issue, we can invoke the upper and lower bounds of $h_i(\theta, \lambda)$ and conduct the grid sampling to efficiently construct the coreset.

## 5.2 Regression

For the regression problem, $\mathbb{Y} = \mathbb{R}$ and the loss function $\ell(\theta, \xi) = L(\theta^\top x - y)$, where $L(\cdot)$ is a non-negative function. Let $\|\cdot\|_*$ be the dual norm of $\|\cdot\|$ on $\mathbb{R}^{m+1}$. We consider the **robust regression** problem that takes the Huber loss $L(z) = \frac{1}{2}z^2$ if $|z| \leq \delta$ and $L(z) = \delta\left(|z| - \frac{1}{2}\delta\right)$ otherwise for some $\delta \geq 0$. If $p = 1$ and $\mathbb{X} = \mathbb{R}^m$, by the result of Shafieezadeh-Abadeh et al. [46, Theorem 3.1], we have

- $R = 0$, $\kappa(\theta) = \mathtt{C}(\theta) = \delta\|(\theta, -1)\|_*$;
- $a_i(\theta, \lambda) = b_i(\theta, \lambda) = h_i(\theta, \lambda) = L(\theta^\top x_i - y_i)$ for any $\lambda \geq \kappa(\theta)$.

## 6 Experiments

Our experiments were conducted on a server equipped with 2.4GHZ Intel CPUs and 256GB main memory. The algorithms are implemented in Python. We use the MOSEK [2] to solve the tractable reformulations of WDROs. Our code is available at `https://github.com/h305142/WDRO_coreset`.

**Compared methods.** We compare our dual coreset method DUALCORE with the uniform sampling approach UNISAMP, the importance sampling approach IMPSAMP [51], the layer sampling approach LAYERSAMP [23], and the approach that directly runs on the whole training set WHOLE.

**Datasets.** We test the algorithms for the SVM and logistic regression problems on two real datasets: MNIST[28] and LETTER[8]. To simulate the scenarios with contaminated datasets, we perform poisoning attacks to the training set of LETTER. Specifically, we use the MIN-MAX attack from [26] and ALFA attack from [54]. We add the standard Gaussian noise $\mathcal{N}(0, 1)$ to the training set of MNIST and randomly flip $10\%$ of the labels. The dual coreset algorithm for the robust regression problem is evaluated on the real dataset APPLIANCES ENERGY[7].

**Results.** Let $s$ and $n$ be the coreset size and the training set size, respectively. We set $c := \frac{s}{n}$ to indicate the compression rate and fix the parameter $\gamma = 7$ for all the instances [5] (recall that $\gamma$ is used for defining the distance $\mathtt{d}(\xi_i, \xi_j) = \|x_i - x_j\| + \frac{\gamma}{2}|y_i - y_j|$). The experiments of each instance were repeated by 50 independent trials. We report the obtained worst-case risk $R_{\sigma,p}^{\mathbb{P}_n}(\theta_*)$ for each method in table 1, 2 and 3. Due to the space limit, the detailed experimental results are placed to the full version of this paper [25].

## 7 Conclusion

In this paper, we consider reducing the high computational complexity of WDRO via the coreset method. We relate the coreset to its dual coreset by using the strong duality property of WDRO, and propose a novel grid sampling approach for the construction. To the best of our knowledge, our work is the first systematically study on the coreset of WDRO problems in theory. We also implement our proposed coreset algorithm and conduct the experiments to evaluate its performance for several WDRO problems (including the applications mentioned in Section 5). Following our work, there also exist several important problems deserving to study in future. For example, it is interesting to consider the coresets construction for other robust optimization models (e.g., adversarial training [20]).

---

[5]We let $\gamma = 7$, which was same as the value set in [31]. We refer readers for a detailed discussion on $\gamma$ to the full version of this paper [25].

| $c$ | UNISAMP | IMPSAMP | LAYERSAMP | DUALCORE |
|---|---|---|---|---|
| 1% | 0.72518±0.1268 | 0.70484±0.0915 | 0.70988±0.084 | **0.68933±0.0587** |
| 2% | 0.64630±0.0344 | 0.65798±0.0447 | 0.63911±0.0305 | **0.63708±0.0257** |
| 3% | 0.62709±0.0216 | 0.63015±0.0333 | **0.62381±0.0199** | 0.62546±0.0225 |
| 4% | 0.62047±0.0176 | 0.6235±0.0183 | 0.61616±0.0143 | **0.61292±0.0149** |
| 5% | 0.61338±0.0164 | 0.61524±0.0137 | 0.61013±0.0096 | **0.60986±0.0097** |
| 6% | 0.60823±0.0084 | 0.61284±0.0131 | 0.60749±0.0119 | **0.60556±0.0092** |
| 7% | 0.60716±0.0082 | 0.61198±0.0113 | 0.6059±0.0083 | **0.60381±0.0073** |
| 8% | 0.60640±0.007 | 0.60936±0.0108 | 0.60376±0.0078 | **0.60238±0.0062** |
| 9% | 0.60395±0.0066 | 0.60677±0.0086 | 0.60235±0.007 | **0.60056±0.0046** |
| 10% | 0.60220±0.0069 | 0.60574±0.009 | **0.60007±0.0041** | 0.60113±0.0071 |

Table 1: Worst-case risk of logistic regression on LETTER, WHOLE=0.59267, $\sigma = 0.3$

| $c$ | UNISAMP | IMPSAMP | LAYERSAMP | DUALCORE |
|---|---|---|---|---|
| 1% | 0.68707±0.1094 | **0.66576±0.103** | 0.70577±0.1278 | 0.67866±0.1173 |
| 2% | 0.59376±0.0565 | 0.60895±0.0683 | **0.58967±0.0529** | 0.59850±0.0548 |
| 3% | 0.56860±0.036 | 0.57346±0.0453 | 0.56705±0.0377 | **0.56689±0.0347** |
| 4% | 0.54429±0.0308 | 0.55050±0.0409 | 0.54366±0.0336 | **0.53634±0.0207** |
| 5% | 0.53218±0.0234 | 0.54212±0.0295 | **0.52981±0.0182** | 0.53217±0.019 |
| 6% | 0.5346±0.0248 | 0.53835±0.0288 | **0.52496±0.0177** | 0.52835±0.0184 |
| 7% | 0.52784±0.0225 | 0.53388±0.0275 | 0.52039±0.015 | **0.52025±0.0147** |
| 8% | 0.52246±0.019 | 0.51993±0.0119 | 0.51918±0.0126 | **0.51845±0.0116** |
| 9% | 0.52025±0.0153 | 0.52402±0.0206 | 0.51289±0.0094 | **0.51196±0.0054** |
| 10% | 0.51458±0.0083 | 0.51768±0.0166 | 0.51578±0.013 | **0.51066±0.0065** |

Table 2: Worst-case risk of SVM on LETTER, WHOLE=0.49734, $\sigma = 0.1$

| $c$ | UNISAMP | IMPSAMP | DUALCORE |
|---|---|---|---|
| 1% | 28.57655±0.0005 | 28.57649±0.0004 | **28.57627±0.0002** |
| 2% | 28.57619±0.0001 | 28.57617±0.0001 | **28.57607±0.0001** |
| 3% | 28.57609±0.0001 | 28.5761±0.0001 | **28.57598±0.0001** |
| 4% | 28.57600±0.0001 | 28.57601±0.0001 | **28.57593±0.0001** |
| 5% | 28.57595±0 | 28.57598±0.0001 | **28.57588±0** |
| 6% | 28.57593±0 | 28.57594±0.0001 | **28.57587±0** |
| 7% | 28.57591±0 | 28.57592±0.0001 | **28.57585±0** |
| 8% | 28.57589±0 | 28.57589±0 | **28.57584±0** |
| 9% | 28.57589±0 | 28.57589±0 | **28.57583±0** |
| 10% | 28.57588±0 | 28.57588±0 | **28.57582±0** |

Table 3: Worst-case risk of Huber regression on APPLIANCES ENERGY, WHOLE=28.57578, $\sigma = 100$. LAYERSAMP coinsides with DUALCORE for Huber regression. This is because $h(\theta, \lambda, \xi_i) = \ell(\theta, \xi_i)$ for Huber regression (See Section 5.2).

# 8 Acknowledgements

The research of this work was supported in part by National Key R&D program of China through grant 2021YFA1000900 and the Provincial NSF of Anhui through grant 2208085MF163. We also want to thank the anonymous reviewers for their helpful comments.

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
