# OpenReview forum: "Coresets for Wasserstein Distributionally Robust Optimization Problems"
_NeurIPS.cc/2022/Conference — NeurIPS 2022 Accept_

### Official Review · Reviewer_hEQP · 2022-07-11

**Rating:** 5
**Confidence:** 3
**Soundness:** 2 fair
**Presentation:** 2 fair
**Contribution:** 2 fair

**Summary:**

The paper proposes a method to reduce the complexity of Wasserstein distributionally robust optimization (WDRO) in machine learning via coresets. Instead of considering the full structure of the Wasserstein ball for the worst-case distribution, the paper approximates this worst-case distribution by a coreset. The authors then propose an implementation for this idea by formulating the “dual coreset” for the dual of the WDRO problem and constructing it empirically using a grid partition framework.


-------- Update: following the discussion at the rebuttal phase, I increased my score evaluation to 5.

**Questions:**

1. I doubt if ``coreset” is the correct terminology that we would like to study in this setting. In my limited understanding, a coreset should be task-agnostic and model-agnostic. However, the coreset in this paper is both task-dependent and task-agnostic, moreover this coreset also depends on the size of the ambiguity set $\theta$. This raises the following issues:

If we change the model (say, from logistic regression to SVM),  then we need to find the coreset again.
If we change the radius of the Wasserstein ambiguity set, then we need to find the coreset again.

This seems to be very inefficient for MLOps because trying different models and tuning the radius are what we do everyday. Rerunning the coreset algorithm of this paper everytime we make tiny changes to the workflow is counterproductive.

[I think this paper is more similar to the idea of data distillation rather than coreset]

2. It is not clear how the sparsity of $W$ is imposed by the algorithm. Can the authors please elaborate on this sparsity?

3. A practical problem is as follows: Assume that I need to train a Wasserstein DRO logistic regression with radius $\theta = 1$, and I have one million samples. I want to find a coreset with one thousand samples, how can I use the results of this paper to accomplish this?

4. I believe that the main paper has to be self-contained, and the reviewers are not obliged to read the appendix. However, the authors are relegating the numerical experiments to the appendix, and the main paper is ending abruptly.

5. The demonstration for applications is severely limited. Particularly, the authors only evaluate the application in adversarially robust binary classification over MNIST and LETTER, and the application in adversarially robust regression over Appliances Energy. The effectiveness of the method in other relevant tasks such as out-of-distribution robustness, worst-group robustness, etc., is not discussed. The reported results only compare the proposed method with a strawman method of UNISAMP, instead of other established WDRO methods. The details for the UNISAMP are also not clear, to which I assume the uniform sampling is an alternative approach to constructing the coreset in place of the proposed grid partition.


Minor comments:
- Some drawings and visualizations may help explain the algorithm more clearly

**Limitations:**

Maybe a few lines of societal impact may be useful to the reader, e.g.: is the coreset method sensitive to (adversarial/poison) attacks?

**Strengths And Weaknesses:**

Originality: The paper appears to be the first to employ the coreset treatment for the worst-case formulation in WDRO problems. The numerical routine for coreset construction seems to be an extension of previous methods [1, 2].

Clarity: A severe clarity issue with the paper is the lack of experimental results in the main paper. The evaluation context, and the competitive edge of the method are also not made clear. The proof section could be hard to follow without the re-stated theorems and the placement of Theorem 1 does not follow the order of introduction in the main paper.  Because the clarity of the paper is poor, I still have difficulty in judging the originality and the true significance of this paper. [Further comments in the next section]

Significance: While the paper is the first to solve WDRO with the coreset treatment, the effectiveness of this approach is not clear due to the limited experiments. Thus, I doubt the significance of the paper in advancing our state of the art.

---

> ### Author Response · Authors · 2022-08-02
> **Response to reviewer hEQP**
>
> Thanks for your thoughtful comments and questions.
> ### Reply to question 1:
> >I doubt if ``coreset” is the correct terminology that we would like to study in this setting. In my limited understanding, a coreset should be task-agnostic and model-agnostic.
> >This raises the following issues:
> If we change the model (say, from logistic regression to SVM), then we need to find the coreset again.
>
> To our best knowledge, most of existing coreset techniques are not task-agnostic and model-agnostic (please refer to the recent surveys on coresets [12, 31]). For example, the popular sensitive-sampling based coreset method [12] (line 98-102) can be applied to various machine learning models (e.g., k-means clustering and logistic regression), however, different models need different construction algorithms and analyses (e.g., k-means and logistic regression have different pseudo-dimensions and different computations on the sensitivities).
> >......moreover this coreset also depends on the size of the ambiguity set $\theta$. If we change the radius of the Wasserstein ambiguity set, then we need to find the coreset again.
>
> Thanks for this question. It is true that our coreset depends on the given parameter $\theta$. But we can avoid frequently updating our coreset by using the simple doubling technique. For example, suppose we already have the coreset for $\theta=\theta_0$, and we want to tune the parameter $\theta$ gradually within a range. When $\theta$ increases and exceeds $2\theta_0$, we can construct the coreset of  $2\theta_0$ (note an $\epsilon$-coreset for a smaller $\theta$ is also an $\epsilon$-coreset for larger $\theta$s; the larger the parameter $\theta$, the smaller the coreset size); if $\theta$ exceeds $4\theta_0$, we can construct the coreset of $4\theta_0$, and so on and so forth. Similarly, if $\theta$ decreases, we can try $\theta_0/2, \theta_0/4, \cdots$.
> ### Reply to question 2 and question 3:
> >It is not clear how the sparsity of $W$ is imposed by the algorithm. Can the authors please elaborate on this sparsity?
>
> >A practical problem is as follows: Assume that I need to train a Wasserstein DRO logistic regression with radius $\theta=1$, and I have one million samples. I want to find a coreset with one thousand samples, how can I use the results of this paper to accomplish this?
>
> The number of non-zero entries in $W$ is actually the size of the obtained coreset (see Definition 1). In our Algorithm 1, we do not input the size of coreset; instead, we only have the pre-specified error bound $\epsilon$. The relation of the coreset size and the error bound is analyzed in Section 4.2. Given the error bound $\epsilon$, in Theorem 2 we show that how large the obtained coreset is from Algorithm 1.
>
> For question 3, if we want to specify the size before constructing the coreset, we can slightly modify our algorithm. Given the sample budget (i.e., the pre-specified coreset size) $1000$, we can perform the grid partition to the dataset as described in Algorithm 1. We compute the sample complexity in each cell according to Lemma 2 (with the constants analyzed in line 255 and line 271). Then sample $1000$ data points proportionally in all cells.
>
>
>
>
>
> ### Reply to question 4:
> >I believe that the main paper has to be self-contained, and the reviewers are not obliged to read the appendix. However, the authors are relegating the numerical experiments to the appendix, and the main paper is ending abruptly.
>
> Thanks for the suggestion, and we will modify the paper's structure accordingly.
>
>
> ### Reply to question 5:
> > The effectiveness of the method in other relevant tasks such as out-of-distribution robustness, worst-group robustness, etc., is not discussed.
>
> Thanks for this question. We agree that it is interesting to consider other robust optimization models. Actually, though our proposed coreset construction is designed for the WDRO model, one can also apply it to compress the data for other robust optimization models; the only issue is that the we may need to develop new idea to analyze the quality guarantee in theory.
>
> >The reported results only compare the proposed method with a strawman method of UNISAMP, instead of other established WDRO methods.
>
> Actually our proposed method is the first coreset approach for the WDRO problem (to our best knowledge), so we only compare it with uniform sampling. To respond the reviewer's question, we also consider to add some comparisons with other coreset methods (though they are not designed for WDRO), e.g. https://arxiv.org/abs/2006.05482 (IMPSAMP) and  https://arxiv.org/abs/2112.02504 (LAYERSAMP).
> Please see the main official comments for detailed experimental results.

---

> ### Comment · Reviewer_hEQP · 2022-08-04
> **still need further clarification**
>
> I thank the authors for the replies. The answer are somehow to brief and I still do not get certain points. I appreciate if the authors can elaborate the followings:
>
> 1. If my grid for $\theta$ is $[0.1, 0.21, 0.43, 0.88]$, does it mean that I need to solve the coreset problem 4 times? Somehow I believe that the advantage of coreset is that I can reduce the memory needed to store data (data compression in some sense): instead of storing 1 million samples, I can store only 1000 samples and I still can guarantee that the performance does not degrade too much. But what the authors propose seem to be: storing both datasets (one with 1 million samples, and a coreset with 1000 samples) and resolving each time (from scratch) to get a coreset. This is impractical and inefficient.
>
> 2. Does it exist a bijection (1-to-1) map between $\epsilon$ and the sparsity of $W$ (this map can depend on the problem’s input)? (Theorem 4.2 is only a big-O relationship, not a bijection map)
>
> 3. Can the authors please give some values on the relationship between $\epsilon$ and the sparsity of $W$ for a practical setting (for example, logistic regression, MNIST data set, etc.)?
>
> Thank you!

---

> > ### Author Response · Authors · 2022-08-05
> > **Further clarification**
> >
> > Thanks for your fast response.
> >
> > >But what the authors propose seem to be: storing both datasets (one with 1 million samples, and a coreset with 1000 samples) and resolving each time (from scratch) to get a coreset.
> >
> > Akin to many other coresets, our dual coreset also enjoys the following two properties, which allow it to be efficiently implemented by the ``Merge-and-Reduce" framework (S. Har-Peled and S. Mazumdar. On coresets for k-means and k-median clustering, in STOC 2004.) if the data size is too large and we cannot afford to store it in our memory.
> >
> >
> >
> > (i) **Merge property**: the union of two coresets is still a coreset for the union dataset.
> > Formally, given two disjoint datasets $P$ and $P'$, if $C$ is a dual $\epsilon$-coreset for $P$ and $C'$ is a dual $\epsilon'$-coreset for $P'$, then $\alpha C\cup (1-\alpha)C'$ is a dual $\max\{\epsilon,\epsilon'\}$-coreset for the dataset $P\cup P'$. Here $\alpha=\frac{|P|}{|P|+|P'|}$ and $\alpha C$ is the weighted set where the weight of each element is multiplied by the factor $\alpha$.
> >
> > (ii) **Reduce property**: the coreset of the coreset is also a coreset for the original dataset.
> > Formally, if $C'$ is a dual $\epsilon'$-coreset for the dataset $P$ and $C$ is a dual $\epsilon$-coreset for the dataset $C'$, then $C$ is a dual $(\epsilon+\epsilon'+\epsilon\cdot\epsilon')$-coreset for $P$.
> >
> > Roughly speaking, the ``Merge-and-Reduce" method utilizes a sequence of “buckets” to maintain the coreset for the input (streaming) data. So we do not need to store the whole data in our memory. The buckets are recursively merged by a bottom-up manner and eventually yields a Merge-and-Reduce tree.
> >
> > We appreciate the reviewer for posing this question, and we will add these details to our paper.
> >
> >
> > >Does it exist a bijection (1-to-1) map between $\epsilon$ and the sparsity of $W$ (this map can depend on the problem’s input)? (Theorem 4.2 is only a big-O relationship, not a bijection map)
> >
> >
> >
> > We believe the exact closed form of this bijection is not easy to obtain (it may require the theoretical lower bound for the coreset size). Actually, the lower bound of coreset size is also a challenging problem in theory, e.g., even for the simple k-clustering, the exact lower bound is still not very clear (Baker et al., Coresets for Clustering in Graphs of Bounded Treewidth, in ICML 2020; Cohen-Addad et al., A New Coreset Framework for Clustering, in STOC 2021). Our big-O result is an upper bound for the sparsity, that is, if the error is no larger than $\epsilon$, in the worst case how large the sparsity can be.
> >
> >
> > >Can the authors please give some values on the relationship between $\epsilon$ and the sparsity of $W$ for a practical setting (for example, logistic regression, MNIST data set, etc.)?
> >
> > Sure. We gave some results in the main official comment. Take the WDRO SVM on Letter dataset for example, if the sample size (i.e. the proportion of non-zero entries in $W$) is $10\%$, WHOLE=0.49734 and DUALCORE=0.51066, and hence the corresponding $\epsilon=\frac{0.51066-0.49734}{0.49734}\approx 0.0268$.

---

> > > ### Comment · Reviewer_hEQP · 2022-08-05
> > > **solution time**
> > >
> > > Thanks for the answer!
> > >
> > > What is the total solution time of the proposed method (coreset + Wasserstein DRO using coreset) compared to full training (Wasserstein DRO using full training data)?

---

> > > > ### Author Response · Authors · 2022-08-06
> > > > **Solution time**
> > > >
> > > > Solving the WDRO usually needs the tractable reformulation (mostly convex constrained programming) [41]. The most popular method for the convex constrained programming in practice is the interior-point method, which needs time $O(n^3)$ [6].
> > > >
> > > > We consider two settings for the coreset method.
> > > >
> > > > i).  **Vanilla coreset** (we store the whole dataset in our memory)
> > > >
> > > > The coreset construction time is  $T_{1} = O(n\cdot \mathtt{time_{ab}})$  (recall that $\mathtt{time_{ab}}$ is the time complexity for computing the lower bound and the upper bound for each $h_i$. The computation of these bounds for the applications studied in Section 5 is specified in line 283 and line 295, which implies $\mathtt{time_{ab}}$ is linear in $d$.)
> > > >
> > > > The coreset size (i.e., the sparsity of $W$) is  $\mathtt{nnz}(W)=O(\frac{d}{\epsilon^2}\cdot\log^3 n)$. So solving the WDRO on the coreset needs time $T_2=O(\mathtt{nnz}(W)^3)$.
> > > >
> > > > The total time  is $T_1+T_2$.
> > > >
> > > >
> > > > ii). **Coreset with ``Merge and Reduce" framework**  (only need to store the coresets in our memory)
> > > >
> > > > The coreset construction time is $T_1=O\left((n+\frac{d}{\epsilon}\cdot\log^4 n)\cdot \mathtt{time_{ab}}\right)$ (the extra $\frac{d}{\epsilon}\cdot\log^4 n \cdot \mathtt{time_{ab}}$ is due to the complexity of the Merge-and-Reduce framework.)
> > > >
> > > > The coreset size $\mathtt{nnz}(W) =  O\left(\frac{d}{\epsilon^2}\cdot\log^3 n\cdot\mathtt{poly}\left(\log(\frac{d}{\epsilon}),\log\log n\right)\right)$. So solving the WDRO on the coreset needs time $T_2=O(\mathtt{nnz}(W)^3)$.
> > > >
> > > > The total time is also $T_1+T_2$ as i).

---

> > > > > ### Comment · Reviewer_hEQP · 2022-08-06
> > > > > **sorry, i was asking about the wallclock time**
> > > > >
> > > > > Sorry i was not entirely clear in my question. I was wondering how long (in seconds) does it take to run the coreset algorithm.

---

> > > > > > ### Author Response · Authors · 2022-08-07
> > > > > > **Wallclock time**
> > > > > >
> > > > > > Sure. For example, running our coreset algorithm on a data matrix of size $12000\times 784$ for WDRO logistic regression takes 0.126s in our experiments. The sparsity of the returned $W$ is 0.5%. The total speed up rate (the time for running on the full training data over the time for coreset construction + running on the coreset) is 38. Another example is  a data matrix of size $500000\times 100$, where it takes 1.291s to construct the coreset. The corresponding total speed up rate is 154.

---

> ### Comment · Reviewer_hEQP · 2022-08-08
> **further thoughts**
>
> I thank the authors for their replies.
>
> There are some points that I would like to highlight (not in any particular order):
>
> - I acknowledge that that the paper seems to be the first one that combines coreset ideas for Wasserstein distributionally robust optimal (WDRO) problem. This is considered novel to me.
> - After reading further on the topic of WDRO, it seems to me that the WDRO is more prevalent when the number of samples is small. For large dataset, the empirical risk minimization problem works well. This paper is positioning itself on the large sample size + WDRO, which brings me some concerns about whether the setting is realistic.
> - This paper emphasizes on the computational tractability of the WDRO (line 58-70). Reference [23] (Kuhn et al. 2019) also proposed a method to alleviate the dependence of the problem on the sample size using a moment set (Gelbrich). This Gelbrich approach also leads to some upper bounds of the original problem, but it seems to work only for type-2 Wasserstein set (see also arXiv:2112.09959). So I think the approach proposed by the authors here is more general than the Gelbrich approach.
> - Coupled with my previous comments, it is not clear to me whether there is a real usecase in practice for this paper.
>
> I raised my score to a 5, but deep inside, I believe that there are still many loopholes in this approach.
>
> I would encourage the authors to submit the revisions of the paper (after taking the comments of the reviewers into consideration, adding new results and I hope the authors can also discuss the limitation of their work).

---

> > ### Author Response · Authors · 2022-08-09
> > **The usecase in practice**
> >
> > Thank you for the thoughtful comments.
> >
> > We agree that some existing works on WDRO are for improving the "out-of-sample" performance where  the training set is not large, but we also note that the data-ambiguous problem can be caused by other practical issues like natural data noise, potential adversarial attacks, or the constant changes of the underlying distribution, e.g., continual learning (please see line 22-line 23). And the data set can be large in these scenarios. So we believe that our WDRO coreset method has various applications beyond the ``small-data'' setting.
> >
> > As suggested by the reviewer, we will provide more new results and the discussions on the limitations of our work in our paper.

---

> > > ### Comment · Reviewer_hEQP · 2022-08-09
> > > **unsubstantiated claim**
> > >
> > > the authors need to be careful with the claim that Wasserstein DRO can be used for continual learning. As far as I know, there is no theoretical claim to justify that WDRO can address the constant changes of distribution. The guarantees that WDRO have mainly focus on iid data, see Esfahani and Kuhn (2018). If the authors are aware of papers that provide the theoretical guarantees of WDRO for distributional shift, I would love to learn as well.

---

> > > > ### Author Response · Authors · 2022-08-09
> > > > **About continual learning**
> > > >
> > > > Sorry for the ``unsubstantiated claim’’. But we would like to mention that there exist some studies on the application of  DRO (Distributionally Robust Optimization) for the continual learning (e.g., Wang et al., Improving Task-free Continual Learning by Distributionally Robust Memory Evolution in ICML 2022). Though their DRO is KL-divergence DRO rather than Wasserstein DRO, we believe the relation between WDRO (as a relatively new model of DRO) and continual learning (and the issue of constant changes of distribution) is an interesting problem deserving to study in future.

---

### Official Review · Reviewer_ts4N · 2022-07-11

**Rating:** 6
**Confidence:** 3
**Soundness:** 3 good
**Presentation:** 3 good
**Contribution:** 2 fair

**Summary:**

The paper constructs a unified framework to construct the ϵ-coreset for the general WDRO problems to reduce the high computational complexity of WDRO. To circumvent the complexity of directly constructing coreset for WDRO, the paper uses dual ϵ-coreset and proposes a more practical “grid sampling” framework and evaluates its performance for several WDRO problems.

**Questions:**

The formula symbols in the paper are a little different from the standard formula symbols, it would be easier to understand if they were standardized. And the proof of the paper is relatively complex and abstract, can it be analyzed with some chart and graph?

**Limitations:**

The authors adequately addressed the limitations and potential negative societal impact of their work.

**Strengths And Weaknesses:**

Strengths:
The paper is clearly structured and logically coherent. The purpose of this paper is to optimize the WDRO problem by corset method, which has certain research significance.
The paper simplified the traditional coreset by solving its dual reformulation, which was innovative to a certain extent.
The paper gave a very detailed derivation process for the bound of λ and gave a detailed theoretical analysis of the setting of the algorithm parameters of epsilon1, epsilon2 and epsilon3.

Weaknesses：
The experimental part of the paper was insufficient, and it was recommended to add some comparative experiments with other WDRO methods.

---

> ### Author Response · Authors · 2022-08-02
> **Response to reviewer ts4N**
>
> Thanks for your thoughtful comments and questions.
>
> >The experimental part of the paper was insufficient, and it was recommended to add some comparative experiments with other WDRO methods.
>
> Thanks for the suggestion. The experiment section will be moved to the main paper of our revised version, and more comparative experiments will be conducted. Please see our reply to question 5 for reviewer hEQP.
>
> ## Response to questions
>
> >The formula symbols in the paper are a little different from the standard formula symbols, it would be easier to understand if they were standardized.
>
> Thanks for the question, and we will modify the symbol style.
>
> >And the proof of the paper is relatively complex and abstract, can it be analyzed with some chart and graph?
>
> Thanks for the suggestion, and we will improve our writing and illustrate our proofs with more intuitive figures.

---

### Official Review · Reviewer_TFqK · 2022-07-11

**Rating:** 7
**Confidence:** 3
**Soundness:** 4 excellent
**Presentation:** 2 fair
**Contribution:** 4 excellent

**Summary:**

The paper introduces a method for computing a coreset for Wasserstein distributionally robust optimization (WRDO) problems by looking at the dual WRDO problem and extending the concept of coresets for the dual. Supported by their theoretical analysis, the authors introduce an algorithm which exploits characteristics of the dual which make the coreset computation easier and prove a sample complexity, which holds with probability proportional to the dataset size. Finally, the paper shows how the proposed coresets could be computed for SVMs, logistic regression, and robust regression problems.

**Questions:**

Minor points:
- l. 159, 237: def of $r_i$ missing $p$ as superscript?
- l. 248: is the sentence missing a period or not finished?


**Limitations:**

I believe that for being a theoretical paper, this makes the limitations clearer by nature. The authors provide full theoretical analyses and proofs to the claims made.

**Strengths And Weaknesses:**


**Strengths**

The sound theoretical analysis, clarity of writing, and significance of the contribution all sum up to a strong submission. I also appreciate how the
authors managed to also include experiments employing their coreset algorithm, albeit only in the Appendix.

**Weaknesses**

I believe the major weaknesses in the paper are the presentation and positioning with regards to related work. Especially as someone not deeply familiar with related literature, I found the related work section particularly weak, simply mentioning related problems and not clearly communicating why coresets for WRDO problems fill an important gap left by existing work on similar problems. To be clear, I mean specific citations, not only a motivation discussion as done in the introduction, which was appropriate.

On the presentation side, I believe the problem is on the choice of content to be part of the main paper versus appendix. Some details are somewhat less relevant, for instance Proposition 1, the proof of Lemma 1, full proof of Theorem 2. Considering the wider audience of NeurIPS and that the content would be present in the appendix for the more interested readers, including the experiments in the main paper would have been a better choice to communicate the authors' findings.

---

> ### Author Response · Authors · 2022-08-02
> **Response to reviewer TFqK**
>
> Thanks for your thoughtful comments and questions.
>
> >Some details are somewhat less relevant, for instance Proposition 1, the proof of Lemma 1, full proof of Theorem 2. Considering the wider audience of NeurIPS and that the content would be present in the appendix for the more interested readers, including the experiments in the main paper would have been a better choice to communicate the authors' findings.
>
> Thanks for the suggestions, and we will move the proofs to the appendix.
>
> ## Response to questions:
> >l. 159, 237: def of $r_i$ missing $p$  as superscript?
>
> The superscript $p$ is omitted for simplicity, and we will clarify the abbreviation in our paper.
>
> >l. 248: is the sentence missing a period or not finished?
>
> Yes and thanks for reminding.

---

### Official Review · Reviewer_9nry · 2022-07-12

**Rating:** 7
**Confidence:** 3
**Soundness:** 3 good
**Presentation:** 4 excellent
**Contribution:** 4 excellent

**Summary:**

The authors first define an $\epsilon$-coreset for the Wasserstein DRO problem as the following (modification of the geometric coresets to the underlying setting):  the coreset is a weighted (desirably sparse weights) modification of the empirical distribution of the training set whose worst-case risk would be very close (driven by $\epsilon$) to the worst-case empirical risk for any feasible solution. The same definition is reused by the authors to guarantee a similar bound on the objective value of the dual of nature's optimization problem (the worst-case risk), and such weights give the *dual* coreset, which the authors coin. They then show these two definitions are identical under some mild continuity/smoothness assumptions on the loss function. By exploiting the nature of the dual problem, the authors derive an efficient algorithm to construct a coreset (with high probability). The coreset WDRO approach is shown to outperform some benchmark methods on well-known machine learning datasets.

**Questions:**

- The distance metric used in the Wasserstein ball is typically named the feature-label distance/metric. Perhaps the authors can cite resources to show the reader this is the most common metric used?
- It would be great to see further numerical experiments in the main text, and for space, the proof of Lemma 1 can be moved into the Appendix?
- Can the authors compare their coreset-WDRO logistic regression with the standard one in [40]?
- I am specifically wondering to what extent can the coreset approach generalize the findings in [41] and [11]. For example, can we characterize the worst-case distributions as in the standard-setting? Can we prove finite-sample guarantees? Would the worst-case distributions look different than the non-coreset approaches? I believe the answers will be affirmative, however, it would be interesting to see further connections with the results derived for the pure Wasserstein settings.

Minor comments:
- Please capitalize "wasserstein" in references.
- Please change alignment $\arg \underset{x}{\max}$ to $\underset{x}{\arg \max }$

**Limitations:**

The authors carefully explain the limitations. I do not believe there is an immediate risk of the negative social impact of this work, as it is rather an enhancement technique for an already well-known set of methods.

**Strengths And Weaknesses:**

Strength:
- I believe the paper is solving a very interesting problem. The results and proofs all look correct to me, and the simplicity of the algorithm to construct the dual coreset is positively surprising.

Weakness:
- The numerical experiments look very limited compared to the theoretical richness of the paper. It would be great to see more modern computational experiments with more visualizations. There are some unexplained steps (e.g., why is $\gamma = 7$).

---

> ### Author Response · Authors · 2022-08-02
> **Response to reviewer 9nry**
>
> Thanks for your thoughtful comments and questions.
>
>
>
> >``There are some unexplained steps (e.g., why is $\gamma=7$).''
>
> Actually we did not spend much effort on tuning the parameter $\gamma$, and it seems to have limited influence on the experimental results. We let $\gamma=7$ only because it was set to be $7$ in the paper [26]. Thanks for this question, and we will add some explanation in our paper.
>
>
>
> ### Question 1:
> >The distance metric used in the Wasserstein ball is typically named the feature-label distance/metric. Perhaps the authors can cite resources to show the reader this is the most common metric used?
>
>
>
> Thanks for this question. The feature-label distance has been widely used in the field of machine learning and robust optimization before, such as [23, 41].
>
>
> ### Question 2:
> >It would be great to see further numerical experiments in the main text, and for space, the proof of Lemma 1 can be moved into the Appendix?
>
> Thanks for the suggestions, and we will modify our paper.
>
> ### Question 3:
> >Can the authors compare their coreset-WDRO logistic regression with the standard one in [40]?
>
> Yes, and actually we already have this comparison in the experiment of WDRO logistic regression on MNIST. Please see Table 1 in the appendix, where the column ``WHOLE'' is the results of the standard WDRO logistic regression (i.e., directly run on the whole data).
>
> ### Question 4:
> >I am specifically wondering to what extent can the coreset approach generalize the findings in [41] and [11]. For example, can we characterize the worst-case distributions as in the standard-setting? Can we prove finite-sample guarantees? Would the worst-case distributions look different than the non-coreset approaches? I believe the answers will be affirmative, however, it would be interesting to see further connections with the results derived for the pure Wasserstein settings.
>
> Thanks for posing these interesting questions. We believe the theorems on the worst-case risk in [41] and [11] can be applied to the coreset setting according to our Corollary 1.
>
> The worst-case distribution in the coreset setting still enjoys a finite-support structure though its support set can be different from the one in the standard setting. Moreover, given the coreset, the structure of the worst-case distribution in the coreset setting can still be captured by the ``weighted version" of Theorem 4.4 in [11] and Corollary 2 (iii) in [15].

---

> > ### Comment · Reviewer_9nry · 2022-08-08
> > **Acknowledging the rebuttal**
> >
> > I would like to thank the authors for their response. My questions are more or less resolved. I have read the reviews and rebuttals as well as the paper again, and I am still positive about this work!
> >
> > Some comments about the response:
> > - Can the authors cross-validate $\gamma$? The selection looks arbitrary to me.
> > - About the answer to **Question 1**: I am very familiar with the feature-label metric. I wanted to kindly note that some citations are needed to motivate the selection of such distance.
> > - Follow-up on the answer to **Question 3**: My bad, thank you!
> > - About the answer to **Question 4**: Are the answers supported by knowledge or belief? Can they be proven? Right now the answers sound a little bit vague.
> >
> > Moreover, the points raised by Reviewer hEQP sound very interesting. I am closely following the discussion on *(i)* the need to re-optimize the coreset for different parameters, *(ii)* the terminology and comparison with the literature, *(iii)* sparsity-epsilon tradeoff. Especially the last one would be very interesting to elaborate on.
> >
> > I would like to also thank the authors for the additional numerical results provided.

---

> > > ### Author Response · Authors · 2022-08-09
> > > **Further response**
> > >
> > > Thank you for the thoughtful comments. Here are some discussions on them.
> > >
> > > >Can the authors cross-validate $\gamma$ ? The selection looks arbitrary to me.
> > >
> > > We appreciate for the helpful suggestion, and we will add more discussions on $\gamma$ to our paper. We are now conducting the numerical experiments (not finished yet) with varying the parameter $\gamma$. We will also perform the experiments to study the relation between $\theta$ and the setting of $\gamma$. Actually the value of $\gamma$ and the value of $\theta$ together influence the result of WDRO, since $\gamma$ measures the cost for changing the label and $\theta$ measures the total cost of the distribution shift.  So we believe it is also interesting to study their relation in the experiments.
> > >
> > > > I am very familiar with the feature-label metric. I wanted to kindly note that some citations are needed to motivate the selection of such distance.
> > >
> > > Thanks and we will add extra citations for the feature-label metric.
> > >
> > > >Are the answers supported by knowledge or belief? Can they be proven? Right now the answers sound a little bit vague.
> > >
> > > The answers can be proven by applying the Corollary 2 (i) and (iii) of [15]. The only part we need to pay attention to is that the Corollary 2 have the form "$\frac{1}{n}\sum_{i=1}^n\delta_{\xi_i}$", but our coreset has the form ``$\frac{1}{n}\sum_{i=1}^nw_i\delta_{\xi_i}$’’ (that is, it is a weighted version of  the Corollary 2 of [15]). We believe that the proof for this weighted version is almost as same as the original proof, and we will add the details to our paper.

---

> > > > ### Comment · Reviewer_9nry · 2022-08-09
> > > > **No more questions**
> > > >
> > > > Thank you very much! I do not have further questions.

---

### Author Response · Authors · 2022-08-02
**Additional experimental results**


Here are some experimental results on **worst-case risk** with more comparisons:

https://arxiv.org/abs/2006.05482 (IMPSAMP) and  https://arxiv.org/abs/2112.02504 (LAYERSAMP).

The datasets are described in the previously submitted supplement.

WDRO logistic regression (WHOLE=0.59267):

| Sample Size | UNISAMP        | IMPSAMP     | LAYERSAMP            | DUALCORE               |
| ----------- | -------------- | -------------- | ------------------ | ------------------ |
| 1%          | 0.72518±0.1268 | 0.70484±0.0915 | 0.70988±0.084      | **0.68933±0.0587** |
| 2%          | 0.64630±0.0344 | 0.65798±0.0447 | 0.63911±0.0305     | **0.63708±0.0257** |
| 3%          | 0.62709±0.0216 | 0.63015±0.0333 | **0.62381±0.0199** | 0.62546±0.0225     |
| 4%          | 0.62047±0.0176 | 0.6235±0.0183  | 0.61616±0.0143     | **0.61292±0.0149** |
| 5%          | 0.61338±0.0164 | 0.61524±0.0137 | 0.61013±0.0096     | **0.60986±0.0097** |
| 6%          | 0.60823±0.0084 | 0.61284±0.0131 | 0.60749±0.0119     | **0.60556±0.0092** |
| 7%          | 0.60716±0.0082 | 0.61198±0.0113 | 0.6059±0.0083      | **0.60381±0.0073** |
| 8%          | 0.60640±0.007  | 0.60936±0.0108 | 0.60376±0.0078     | **0.60238±0.0062** |
| 9%          | 0.60395±0.0066 | 0.60677±0.0086 | 0.60235±0.007      | **0.60056±0.0046** |
| 10%         | 0.60220±0.0069 | 0.60574±0.009  | **0.60007±0.0041** | 0.60113±0.0071     |
WDRO SVM (WHOLE=0.49734):

| Sample Size | UNISAMP       | IMPSAMP        | LAYERSAMP            | DUALCORE               |
| ----------- | -------------- | ----------------- | ------------------ | ------------------ |
| 1%          | 0.68707±0.1094 | **0.66576±0.103** | 0.70577±0.1278     | 0.67866±0.1173     |
| 2%          | 0.59376±0.0565 | 0.60895±0.0683    | **0.58967±0.0529** | 0.59850±0.0548     |
| 3%          | 0.56860±0.036  | 0.57346±0.0453    | 0.56705±0.0377     | **0.56689±0.0347** |
| 4%          | 0.54429±0.0308 | 0.55050±0.0409    | 0.54366±0.0336     | **0.53634±0.0207** |
| 5%          | 0.53218±0.0234 | 0.54212±0.0295    | **0.52981±0.0182** | 0.53217±0.019      |
| 6%          | 0.5346±0.0248  | 0.53835±0.0288    | **0.52496±0.0177** | 0.52835±0.0184     |
| 7%          | 0.52784±0.0225 | 0.53388±0.0275    | 0.52039±0.015      | **0.52025±0.0147** |
| 8%          | 0.52246±0.019  | 0.51993±0.0119    | 0.51918±0.0126     | **0.51845±0.0116** |
| 9%          | 0.52025±0.0153 | 0.52402±0.0206    | 0.51289±0.0094     | **0.51196±0.0054** |
| 10%         | 0.51458±0.0083 | 0.51768±0.0166    | 0.51578±0.013      | **0.51066±0.0065** |

WDRO Huber regression (WHOLE=28.57578):


 | Sample Size |UNISAMP        | IMPSAMP      | DUALCORE               |
| ----------- | --------------- | --------------- | ------------------- |
| 1%          | 28.57655±0.0005 | 28.57649±0.0004 | **28.57627±0.0002** |
| 2%          | 28.57619±0.0001 | 28.57617±0.0001 | **28.57607±0.0001** |
| 3%          | 28.57609±0.0001 | 28.57610±0.0001  | **28.57598±0.0001** |
| 4%          | 28.57600±0.0001 | 28.57601±0.0001 | **28.57593±0.0001** |
| 5%          | 28.57595±0      | 28.57598±0.0001 | **28.57588±0**      |
| 6%          | 28.57593±0      | 28.57594±0.0001 | **28.57587±0**      |
| 7%          | 28.57591±0      | 28.57592±0.0001 | **28.57585±0**      |
| 8%          | 28.57589±0      | 28.57589±0      | **28.57584±0**      |
| 9%          | 28.57589±0      | 28.57589±0      | **28.57583±0**      |
| 10%         | 28.57588±0      | 28.57588±0      | **28.57582±0**      |

Remark: LAYERSAMP coinsides with DUALCORE for WDRO Huber regression.

---

### Meta-Review · Area_Chair_5gVJ · 2022-08-26

**Recommendation:** Accept
**Confidence:** Certain

**Metareview:**

From the reviewers' comments and my own reading of the paper, the idea of bringing the notion of coreset from computational geometry to bear on the WDRO problem is novel and has the potential of further development. The authors should carefully address the reviewers' comments in the revision.

**Award:**

No

---

### Decision · Program_Chairs · 2022-09-14

Accept